# Inhibition of Inducible Nitric Oxide Synthase Prevents IL-1β-Induced Mitochondrial Dysfunction in Human Chondrocytes

**DOI:** 10.3390/ijms22052477

**Published:** 2021-03-01

**Authors:** Annett Eitner, Sylvia Müller, Christian König, Arne Wilharm, Rebecca Raab, Gunther O. Hofmann, Thomas Kamradt, Hans-Georg Schaible

**Affiliations:** 1Department of Trauma, Hand and Reconstructive Surgery, Experimental Trauma Surgery, Jena University Hospital, Friedrich-Schiller-University Jena, 07747 Jena, Germany; Arne.Wilharm@med.uni-jena.de (A.W.); Rebecca@raab-online.de (R.R.); Gunther.Hofmann@med.uni-jena.de (G.O.H.); 2Institute of Immunology, Jena University Hospital, Friedrich-Schiller-University Jena, 07743 Jena, Germany; Sylvia.Mueller@med.uni-jena.de (S.M.); Thomas.Kamradt@med.uni-jena.de (T.K.); 3Institute of Physiology 1/Neurophysiology, Jena University Hospital, Friedrich-Schiller-University Jena, 07743 Jena, Germany; Christian.Koenig@med.uni-jena.de (C.K.); Hans-Georg.Schaible@med.uni-jena.de (H.-G.S.); 4Clinic of Trauma, Orthopedic and Septic Surgery, Hospital St. Georg gGmbH, 04129 Leipzig, Germany

**Keywords:** osteoarthritis, NO synthase, Interleukin-1β, chondrocytes, mitochondrial dysfunction

## Abstract

Interleukin (IL)-1β is an important pro-inflammatory cytokine in the progression of osteoarthritis (OA), which impairs mitochondrial function and induces the production of nitric oxide (NO) in chondrocytes. The aim was to investigate if blockade of NO production prevents IL-1β-induced mitochondrial dysfunction in chondrocytes and whether cAMP and AMP-activated protein kinase (AMPK) affects NO production and mitochondrial function. Isolated human OA chondrocytes were stimulated with IL-1β in combination with/without forskolin, L-NIL, AMPK activator or inhibitor. The release of NO, IL-6, PGE_2_, MMP3, and the expression of iNOS were measured by ELISA or Western blot. Parameters of mitochondrial respiration were measured using a seahorse analyzer. IL-1β significantly induced NO release and mitochondrial dysfunction. Inhibition of iNOS by L-NIL prevented IL-1β-induced NO release and mitochondrial dysfunction but not IL-1β-induced release of IL-6, PGE_2_, and MMP3. Enhancement of cAMP by forskolin reduced IL-1β-induced NO release and prevented IL-1β-induced mitochondrial impairment. Activation of AMPK increased IL-1β-induced NO production and the negative impact of IL-1β on mitochondrial respiration, whereas inhibition of AMPK had the opposite effects. NO is critically involved in the IL-1β-induced impairment of mitochondrial respiration in human OA chondrocytes. Increased intracellular cAMP or inhibition of AMPK prevented both IL-1β-induced NO release and mitochondrial dysfunction.

## 1. Introduction

Pro-inflammatory cytokines contribute significantly to the initiation and progression of osteoarthritis (OA) via up-regulation of catabolic processes [1,2]. In addition, the impairment of the mitochondrial function of chondrocytes is thought to be an important factor in the pathophysiology of OA [3,4,5]. Experiments showed that Interleukin-1β (IL-1β) can impair the activity of mitochondrial respiratory chain enzyme complexes [6]. However, the mechanism by which IL-1β modulates mitochondrial respiration remains unclear. IL-1β induces upregulation of inducible NO synthase (iNOS) and the production of nitric oxide (NO). This mediator regulates numerous putative pathogenic processes in the cartilage (see below), and may also alter mitochondrial respiration and ATP production in chondrocytes [3]. However, whether IL-1β-induced NO production is causally responsible for mitochondrial impairment and whether inhibition of iNOS can prevent IL-1β-induced mitochondrial dysfunction has not yet been reported.

An increased iNOS expression was found in OA cartilage and synovial tissue [7]. Although NO produced by constitutive NO synthase at low concentration can reduce OA pain probably through the promotion of blood flow, thus improving oxygen supply and reducing ischemic pain [8], NO produced by the cytoplasmic iNOS at high concentration is a pro-inflammatory factor and contributes to OA pathogenesis [9], and can induce cell damage. In chondrocytes, NO increases the release of pro-inflammatory mediators and inhibits the synthesis of cartilage matrix components and increases the activity of matrix-degrading enzymes such as matrix metalloproteinases (MMP) [10]. Exogenous cytokines can increase iNOS expression and NO release of OA cartilage and cultured chondrocytes [11,12]. In addition, NO at high concentration affects cytochrome c oxidase in mitochondria, induces caspase 3, and is presumably responsible for the initiation of apoptosis [13]. It has been reported that mitochondrial dysfunction also increases the responsiveness of chondrocytes for cytokines [14]. An interesting question is, therefore, whether NO-induced mitochondrial dysfunction also increases the IL-1β-induced release of pro-inflammatory mediators. 

The expression and activity of iNOS are regulated by various signaling pathways such as cyclic adenosine monophosphate (cAMP)- or AMP-activated protein kinase (AMPK)-pathway, and the effect of activation or inhibition of these pathways differs between cell types. It has been reported that the signaling molecule cAMP can increase iNOS expression in adipocytes, smooth muscle cells, and skeletal muscle cells, whereas in hepatocytes and astrocytes cAMP suppressed iNOS expression [15,16]. Tissue-specific gene expression and an alteration of cell signaling pathways are thought to be responsible for these opposite effects of cAMP [16]. Another regulator of iNOS expression is the AMPK, which is an important regulator of cellular energy metabolism. A high AMP/ATP ratio activates AMPK, which inhibits ATP-consuming pathways and increases ATP production [17]. In many cell types, AMPK exerts anti-inflammatory effects and reduces iNOS expression [18]. In contrast, in hepatocytes, activation of AMPK upregulates the cytokine-induced expression of iNOS and NO production [19]. Presumably, the signaling molecule cAMP and the regulator AMPK are part of two independent pathways for the regulation of iNOS expression. Whether cAMP and AMPK modulate IL-1β-induced NO release and mitochondrial function in chondrocytes is unclear.

The current study aimed to evaluate whether increased NO production is responsible for IL-1β-induced mitochondrial dysfunction in human OA chondrocytes obtained from knee joints during arthroplasty. Of particular interest is to evaluate whether blockade of iNOS activity prevents the negative effects of IL-1β on chondrocytes. Additionally, we analyzed the role of cAMP and AMPK in the regulation of NO production and release from these cells and in IL-1β-induced mitochondrial dysfunction. To understand the regulation of iNOS activity in chondrocytes is important to move forward with the development of OA therapies based on iNOS as a target for OA treatment. 

## 2. Results

### 2.1. IL-1β-Induced NO Release

After stimulation with IL-1β for 48 h, cultured chondrocytes showed a concentration-dependent NO release (Figure 1a, repeated measures ANOVA: *p* <0.001). The IL-1β concentration 0.1 ng/mL evoked a high release of NO, which only slightly increased at higher IL-β concentrations (Figure 1a). Therefore, all further experiments were performed with an IL-1β concentration of 0.1 ng/mL. 

Co-application of 0.1 ng/mL IL-1β and 10 µM L-NIL, an inhibitor of iNOS, resulted in a significant reduction of IL-1β-induced NO release (Figure 1b, *p* = 0.006), showing that L-NIL is a useful compound to investigate NO-related effects.

Application of forskolin (50 µM), a cell-permeable activator of adenylyl cyclase increasing the intracellular level of cAMP, caused a significant decrease of IL-1β-induced NO production (Figure 1c, *p* < 0.001, Wilcoxon signed-rank test). The reduction of IL-1β-induced NO release by forskolin was on average to 62.7% (Figure 1d). Application of 8-Bromo-cAMP or PGE_2_, which also induced an increase of intracellular cAMP, reduced the IL-1β-induced NO release on average to 71.9% or 71.0%, respectively (Figure 1d).

To test if the AMPK was involved in the regulation of IL-1β-induced NO release, co-applications of IL-1β with 10 µM A769662, an activator of AMPK, or of IL-1β with 10 µM dorsomorphin dihydrochloride, an inhibitor of AMPK, were performed. Activation of AMPK with A769662 resulted in a significant increase of IL-1β-induced NO release (*p* < 0.001, Wilcoxon signed-rank test with Bonferroni adjustment, Figure 1e). Vice versa, the AMPK-inhibitor dorsomorphin dihydrochloride completely prevented the IL-1β induced NO release to the level of the unstimulated control (Figure 1e).

We found a small basal release of NO. Forskolin or A769662 alone did not influence NO release (Figure 1f). L-NIL or dorsomorphin dihydrochloride alone slightly reduced the basal release of NO compared with the unstimulated control (Figure 1f, *p* = 0.028, Wilcoxon signed-rank test).

### 2.2. Impact on iNOS Protein Expression

Stimulation with 0.1 ng/mL IL-1β caused a significant upregulation of iNOS protein expression (Figure 1g–h, *p* = 0.028). Co-stimulation of IL-1β with the AMPK activator A769662 resulted in a strong increase of IL-1β-induced iNOS expression compared to IL-1β alone (Figure 1g–h, *p* = 0.028). Co-stimulation of IL-1β with the AMPK inhibitor dorsomorphin dihydrochloride resulted in a low basal iNOS expression comparable to the level of unstimulated cells (Figure 1g–h).

### 2.3. Impact of IL-1β on Mitochondrial Function

An application of 0.1 ng/mL IL-1β for 24 h resulted in a strong reduction of mitochondrial basal and maximal respiration as well as ATP production (Figure 2a, Wilcoxon signed-rank test with Bonferroni adjustment: All *p* < 0.001), and a significant increase of non-mitochondrial respiration (*p* = 0.014, Wilcoxon signed-rank test with Bonferroni adjustment, Figure 2a). After IL-1β stimulation, the basal mitochondrial respiration decreased to 61.2%, the maximal mitochondrial respiration to 53.4%, and the ATP production to 50.8% of the unstimulated control. However, the non-mitochondrial respiration increased to 128% of the unstimulated control.

Co-application of 0.1 ng/mL IL-1β and 10 µM L-NIL resulted in a complete recovery of mitochondrial function (Figure 2a). The mitochondrial basal respiration, maximal respiration, and ATP production of chondrocytes stimulated with IL-1β and L-NIL simultaneously significantly increased compared with the IL-1β-stimulated cells (*p* < 0.001, Wilcoxon signed-rank test with Bonferroni adjustment) and did not differ significantly from unstimulated cells. 

Forskolin also prevented the IL-1β-induced impairment of mitochondrial function (Figure 2a). The values of the mitochondrial basal respiration, maximal respiration, and ATP production were not significantly different after simultaneous application of IL-1β and forskolin compared with unstimulated chondrocytes (Figure 2a). Only the IL-1β-induced increase of non-mitochondrial respiration remained elevated (*p* < 0.001, Wilcoxon signed-rank test with Bonferroni adjustment).

Forskolin alone did not show any influence on the mitochondrial or the non-mitochondrial respiration (Figure 2b). An application of L-NIL alone resulted in a slight but significant reduction of basal respiration (93.6% of unstimulated control, *p* = 0.024, Wilcoxon signed-rank test with Bonferroni adjustment) and maximal respiration (95.8% of unstimulated control, *p* = 0.004, Wilcoxon signed-rank test with Bonferroni adjustment). The ATP production and non-mitochondrial respiration were comparable to unstimulated cells.

Activation of AMPK with A769662 aggravated IL-1β-induced mitochondrial dysfunction (Figure 2c). Co-application of IL-1β and A769662 resulted in an additional reduction of basal mitochondrial respiration to 66.4% (*p* = 0.015), of maximal mitochondrial respiration to 26.0% (*p* < 0.001), and of ATP production to 39.6% (*p* = 0.003) of IL-1β stimulation alone (all Wilcoxon signed-rank test with Bonferroni adjustment, Figure 2c). The non-mitochondrial respiration increased to 181% of IL-1β-stimulated cells after co-stimulation with IL-1β and A769662 (*p* < 0.001, Wilcoxon signed-rank test with Bonferroni adjustment, Figure 2c). Compared to the unstimulated control, co-application of IL-1β and A769662 reduced the basal mitochondrial respiration to 41.3%, whereas A769662 alone only reduced the basal mitochondrial respiration to 90.6% of the unstimulated control (Figure 2c,d).

Co-application of IL-1β and the AMPK-inhibitor dorsomorphin dihydrochloride resulted in a significant improvement of IL-1β induced mitochondrial dysfunction (Figure 2c). The basal mitochondrial respiration increased to 139% (*p* = 0.0059), the maximal mitochondrial respiration to 153% (*p* < 0.001), and the ATP production to 178% (*p* = 0.0059) of IL-1β stimulation alone (all Wilcoxon signed-rank test with Bonferroni adjustment, Figure 2c). Compared to the unstimulated control, basal mitochondrial respiration, ATP production, and non-mitochondrial respiration were not significantly different after co-application of IL-1β and dorsomorphin dihydrochloride (Figure 2c), even though dorsomorphin dihydrochloride alone caused a significant reduction of basal (73.8%) and maximal mitochondrial respiration (72.4%) as well as ATP production (72.6%) (Figure 2d).

### 2.4. Effect of iNOS Inhibition by L-NIL on Other Mediators

To test whether the iNOS inhibitor L-NIL also affects IL-1β-induced production of other mediators, the release of IL-6, PGE_2_, and MMP-3 was measured after co-stimulation of IL-1β and 10 µM L-NIL. Upon IL-1β stimulation, L-NIL had only minor effects on the release of IL-6, PGE_2_, and MMP-3 (Figure 3a–c). In addition, the production and release of the cartilage matrix protein glycosaminoglycan were not affected by 10 µM L-NIL (Figure 3d). These data suggest that the effect of IL-1β on the release of IL-6, PGE_2_, MMP3, and GAG did not involve NO.

### 2.5. Impact of Mediators on Vitality of Chondrocytes

To control whether the observed effects were based on cytotoxic effects of the mediators used, an assay was performed to determine the percentage of living and dead chondrocytes after stimulation with different mediators. The percentage of dead cells did not change significantly after stimulation with IL-1β, forskolin, L-NIL, A769662, or dorsomorphin dihydrochloride (Figure 4). After stimulation with IL-1β and forskolin, the viability of chondrocytes was similar to the unstimulated control. L-NIL, A769662, and dorsomorphin dihydrochloride reduced the percentage of living cells slightly, but the observed effect of these mediators on NO release and mitochondrial function cannot be explained by these small effects on the viability. The cytotoxic dimethyl sulfoxide (DMSO) control (1:10) proved the validity of the test, while DMSO 1:200 had no effect.

## 3. Discussion

The results of the current study provide evidence that increased NO production in human chondrocytes is responsible for IL-1β-induced mitochondrial impairment. Inhibition of iNOS by L-NIL prevented the IL-1β-induced reduction of mitochondrial respiration and ATP production. The application or induction of cAMP reduced the IL-1β-induced NO release and impairment of mitochondrial function. Furthermore, the results show that AMPK is an important regulator for the IL-1β-induced NO production in chondrocytes. The activation of AMPK increased the IL-β-induced NO release and, therefore, increased the negative impact of IL-1β on mitochondrial function. The inhibition of AMPK resulted in a strong reduction of NO release and prevented IL-1β-induced impairment of mitochondrial respiration.

Previous studies showed that both IL-1β and NO impaired the activity of respiratory chain enzyme complexes and thus affect mitochondrial function in normal human chondrocytes [6,20]. Therefore, it was supposed that IL-1β-induced NO production may be responsible for the IL-1β-induced mitochondrial dysfunction. Our data now reveal that the effects of IL-1β and NO on mitochondrial function are causally linked. First, the effects of IL-1β on NO release, mitochondrial respiration, and ATP production were prevented by the iNOS inhibitor L-NIL. Second, IL-1β enhanced iNOS expression and NO release in chondrocytes. Third, downregulation of IL-1β-induced NO release by cAMP or the AMPK inhibitor dorsomorphin dihydrochloride was accompanied by a significant improvement of IL-1β-induced mitochondrial impairment.

NO production and iNOS expression were shown to be modulated by cAMP, which activates protein kinases and regulates gene transcription via transcription factors such as CREB. Interestingly, it seems to depend on the cell type whether cAMP increases or decreases iNOS expression and NO production. In our experiment, the stimulation of cAMP synthesis in chondrocytes decreased IL-1β-induced NO production, similar as in hepatocytes and astrocytes, but in cardiac myocytes, macrophages, and vascular smooth muscle cells, NO production was enhanced by cAMP elevation [15,21]. The decrease of NO production by cAMP in hepatocytes and astrocytes was explained by decreasing iNOS mRNA expression, iNOS protein expression, and alteration of iNOS promoter activity. Here, we show for the first time that stimulation of cAMP production by forskolin prevents mitochondrial dysfunction induced by IL-1β, similar to iNOS inhibition, suggesting that the increase of cAMP protects against the negative effect of IL-1β on mitochondrial function by iNOS inhibition or reduced iNOS expression. Since cAMP elevation may also be caused by inflammatory mediators such as PGE_2_, this mechanism may limit the mitochondrial impairment of IL-1β in the inflammatory setting.

Activation of AMPK can also influence iNOS activity. AMPK is an important energy-sensing molecule that can switch off ATP-consuming pathways and switch on pathways for ATP production. AMPK is activated by a high AMP/ATP ratio. The effect of AMPK on iNOS activation also depends on the cell type. In myocytes, adipocytes, and macrophages, pharmacological activation of AMPK significantly inhibited iNOS under pro-inflammatory conditions, primarily resulting from post-transcriptional regulation of the iNOS protein [22]. In hepatocytes, however, AMPK activation increased cytokine-induced iNOS expression and NO production [19]. In human chondrocytes, we found a significant increase of IL-1β-induced NO production and iNOS expression after AMPK activation by A769662, and a marked reduction of IL-1β-induced NO production and iNOS expression after AMPK inhibition by dorsomorphin dihydrochloride, thus resembling the effects in hepatocytes. Additionally, our data show that A769662 aggravates IL-1β-induced mitochondrial impairment, whereas dorsomorphin dihydrochloride prevents IL-1β-induced mitochondrial effects. Since the effect of A769662 on NO production and mitochondrial function in chondrocytes was only observed in combination with IL-1β, the mechanism by which AMPK regulates iNOS should be related to the IL-1β pathways. In primary hepatocytes, AMPK affected cytokine-induced NO production and iNOS expression through Akt, c-Jun N-terminal kinase, and NF-kB signaling pathways [19]. Given that the effects of AMPK in hepatocytes and chondrocytes seem to be similar, we assumed analogous regulatory pathways in chondrocytes. IL-1β reduces mitochondrial ATP production through NO production, therefore, it should activate the cytosolic AMPK. As activation of AMPK increased the IL-1β-mediated NO release in chondrocytes, a vicious circle may be produced if AMPK is not inhibited by other mechanisms. Such a mechanism could be the inhibition of phosphodiesterase (PDE), which hydrolyzes cAMP to AMP. Inhibition of AMP production by PDE inhibitors leads to reduced AMPK activity. In human OA chondrocytes, Tenor et al. found that inhibition of PDE4 decreased IL-1β-induced NO production by reducing iNOS protein expression [23], thus resembling the effect of AMPK inhibition observed in the present study. Importantly, our data show that the modulation of NO synthesis by AMPK signaling also affects IL-1β-induced mitochondrial impairment.

In addition to mitochondrial respiration, AMPK regulates glucose uptake and ROS production, protein synthesis, promotor activity, or receptor activity. Thus the AMPK mechanisms in chondrocytes should be further investigated to evaluate the impact on OA mechanisms.

Since chondrocytes mainly utilize glycolysis for ATP production, which does not involve mitochondrial activity, the relative importance of oxidative phosphorylation for ATP supply in chondrocytes has been discussed. However, several studies reported that mitochondrial dysfunction is involved in pathophysiological processes, which include oxidative stress, apoptosis, production of inflammatory mediators, matrix catabolism, and calcification of cartilage matrix [24]. Inhibition of mitochondrial respiratory complexes increased the expression of cyclooxygenase 2 and the level of PGE_2_ in normal human chondrocytes [25] as well as the inflammatory responsiveness to cytokines [14]. Furthermore, mitochondrial dysfunction induced apoptosis by inducing ROS and mtDNA damage [24]. Thus, intact mitochondrial respiration is thought to be crucial for the homeostasis and survival of chondrocytes. Since NO induces mitochondrial dysfunction, the regulation of NO production and iNOS expression is of particular interest in this context.

While we identified significant negative NO effects on mitochondrial function, our data did not provide evidence that NO is critically involved in the IL-1β-induced production and release of IL-6, PGE_2_, and MMP3. Besides the fact that PGE_2_, IL-6, and MMP3 are important molecules in OA processes, several studies found that mitochondrial dysfunction affects the production of these mediators [14,24,25,26]. In human chondrocytes, mitochondrial dysfunction induced by inhibitors of mitochondrial complexes increased the production of PGE_2_ and MMP3 [25,26] and the inflammatory response to IL-1β [14]. According to these studies, we expected a reduced release of IL-6, PGE_2_, and MMP3 after incubation with L-NIL and prevention of NO-induced mitochondrial impairment. However, we found that the mediators PGE_2_, IL-6, and MMP3 were significantly more released upon stimulation with IL-1β, but the IL-1β-induced release was not affected by L-NIL at concentrations that inhibited the effects of IL-1β on NO production, mitochondrial respiration, and ATP production. Thus, while IL-1β alone reduced mitochondrial function in a NO-dependent manner, this effect was not crucial for the IL-1β-induced release of IL-6, PGE_2_, and MMP3 in our experiments. In the study of Vaamonde-Garcia et al., mitochondrial impairment induced by oligomycin alone already enhanced the basal release of pro-inflammatory mediators, oligomycin combined with IL-1β led to an additional increase of these mediators [14]. Thus the mitochondrial dysfunction was directly initiated by inhibitors additionally to the IL-1β-induced mitochondrial impairment. It seems that mitochondrial dysfunction induced by inhibitors of mitochondrial respiration chain complexes results in a slightly different reaction compared to the inhibitory effect of NO on mitochondrial respiration. In addition, anti-inflammatory effects of NO in chondrocytes were described in the literature. In one study, inhibition of NO synthesis enhanced the IL-1β-induced IL-6 and PGE_2_ production [27]. Concerning the different mentioned effects of NO and inhibitors, several pathways might converge on mitochondrial respiration and result in different responsiveness to cytokines. 

Some limitations of this study should be noted. We did not observe the negative effects of IL-1β on the vitality of chondrocytes upon exposure to IL-1β for two days. Thus mitochondrial dysfunction by IL-1β may not cause rapid cell death. Our study showed putative mechanisms that may protect against IL-1β-induced mitochondrial malfunction. Induction of apoptosis and cytotoxic effects may be observed after a longer stimulation period or higher IL-1β concentrations. Concerning modulation of mitochondrial function by cAMP and AMPK, most experiments were performed with a single concentration of the used mediators according to preliminary experiments and literature. The described effects of cAMP and AMPK could be stronger or weaker with other concentrations or incubation times. In general, in-vitro models presumably do not reflect exactly the in-vivo situation in OA cartilage, but chondrocytes cultured in the monolayer are the most widely used in-vitro model to study the effect of cytokines on molecular pathways of chondrocytes [28].

In summary, our data demonstrate the importance of NO for the IL-1β-induced negative effects on the mitochondrial function in chondrocytes. It supports the concept that inhibition of iNOS could be a beneficial treatment of OA. Treatment with iNOS inhibitors showed chondroprotective effects in animal and human studies [10] and significantly reduced OA progression in an experimental animal model [29]. The chondroprotective effects of iNOS inhibition may result at least in part from the reduction of mitochondrial dysfunction induced by IL-1β.

## 4. Materials and Methods

### 4.1. Reagents/Solutions

Human IL-1β was purchased from PeproTech (Rocky Hill, NJ, USA). N^6^-(1-iminoethyl)-L-lysine hydrochloride (L-NIL), forskolin, 8-Bromoadenosine-3′,5′-cyclic monophosphate sodium salt (8-Bromo-cAMP), dorsomorphin dihydrochloride, and A769662 were all purchased from Tocris Bioscience (Bristol, UK). Forskolin and A769662 were dissolved in DMSO/water (final DMSO dilution 1:200 and 1:5000, respectively), all other substances were dissolved in water. PGE_2_ was from Cayman Chemical (Ann Arbor, MI, USA) and was dissolved in DMSO/water (final DMSO dilution 1:5000). Pronase E was obtained from Merck KGaA (Darmstadt, Germany), and collagenase P from Roche Diagnostics GmbH (Mannheim, Germany). The chondrocytes culture medium contains Chondrocyte Basal Medium + 10% Chondrocyte Growth Medium SupplementMix (both from PromoCell GmbH, Heidelberg, Germany) + 1% Penicillin/Streptomycin Solution (Life Technologies Europe BV, NN Bleiswijk, Netherlands).

### 4.2. Isolation of Human Chondrocytes

Human chondrocytes were obtained from 37 patients (16 female/21 male) with end-stage knee OA who underwent knee replacement surgery. Patients were on average 63.6 years old (±10.1 years, standard deviation). Patients were informed about the purpose of tissue sampling and gave written consent after the nature of all examinations was fully explained. The study was approved by the Ethical Committee for Clinical Trials of the Friedrich-Schiller-University of Jena (ethic approval code: 3966-12/13; date of approval: 23 January 2014) and performed in accordance with the Declaration of Helsinki.

Directly after surgical removal of the condyles, cartilage was removed from the condyles and was cut into small pieces. Cartilage was treated with 0.01 mg/mL pronase E in Dulbecco’s modified Eagles’s medium (DMEM) for 30 min at 37 °C following collagenase P (1.3 mg/mL in chondrocyte culture medium) for 16 h at 37 °C. The cells were filtrated, washed, and seeded in cell culture plates.

### 4.3. Experiments on Release of Mediators

For release experiments, the chondrocytes were plated on 24-well culture plates at a density of 4 × 10^5^ cells/cm^2^ and cultured in the chondrocyte culture medium. After 3 days of incubation, the medium was renewed. After an additional 2 days, the cells were stimulated with IL-1β (10–0.01 ng/mL) and the iNOS inhibitor L-NIL (1, 10, and 20 µM) at different concentrations for 48 h to determine the lowest efficient concentration. After preliminary experiments with different concentrations of all mediators in combination with IL-1β, we stimulated the chondrocytes in the final experiments with 50 µM forskolin, 10 µg/mL PGE_2_, 50 µM 8-Bromo-cAMP, 10 µM L-NIL, 10 µM A769662 (an activator of AMPK), and 10 µM dorsomorphin dihydrochloride (an inhibitor of AMPK) alone or in combination with IL-1β for 48 h. The supernatant of all experiments was collected and stored at −80 °C until use. The cells were plated in duplicate for each condition. Experiments were performed with a minimum of 5 biological replicates (donors) to ensure reproducibility.

Griess assay: Concentration of nitrite in the supernatant was measured using the Griess Reagent Kit (#G7921, Invitrogen, Thermo Fisher Scientific Inc., Darmstadt, Germany,) according to manufacturer’s instruction and used as an indicator for NO synthesis of the cultured chondrocytes.

Measurements of IL-6, PGE_2_, and MMP-3: Concentrations of IL-6, PGE_2_, and MMP-3 in the supernatant were measured by ELISA using IL-6 human uncoated ELISA Kit (#88-7066-22, Invitrogen), Prostaglandin E_2_ ELISA (DRG Instruments, Marburg, Germany, #EIA-5811), and RayBio human MMP-3 ELISA Kit (RayBiotech Inc., Norcross, GA, USA, #ELH-MMP3-5).

Measurement of GAG: The amount of glycosaminoglycan (GAG) was measured spectrophotometrically using 1,9-dimethylmethylene blue (DMB, Sigma-Aldrich, Taufkirchen, Germany). A standard curve of bovine chondroitin sulfate (Sigma-Aldrich) was generated to calculate the GAG concentration.

### 4.4. Vitality of Chondrocytes

Chondrocytes were plated on 96-well culture plates at a density of 4 × 10^5^ cells/cm^2^. The cells were cultured and stimulated as described above. For testing the impact of mediators used on the viability of chondrocytes, the LIVE/DEAD Viability/Cytotoxicity Kit from Invitrogen (#L3224) was performed according to the manufacturer’s instruction. The assay determined the percentage of living and dead cells after stimulation with different mediators.

### 4.5. Measurement of Mitochondrial Function

Functional parameters of mitochondrial respiration were measured with the Seahorse XF Cell Mito Stress Test Kit using the Seahorse XF Analyzer (Agilent, Santa Clara, CA, USA). The assay included modulators of mitochondrial respiration to measure basal respiration, ATP-linked respiration, maximal respiration, and non-mitochondrial respiration. These functional parameters were calculated by measuring the oxygen consumption rate (OCR) of the cultured chondrocytes using the data analysis tool Seahorse XF Report Generator (Agilent). Oligomycin, carbonyl cyanide-4 (trifluoromethoxy) phenylhydrazone (FCCP), and antimycin A/rotenone were sequentially injected to modulate the mitochondrial respiration.

For this purpose, chondrocytes were plated on Seahorse cell culture plates at a density of 3 × 10^4^ cells/well and cultured in chondrocytes culture medium for 6 days at 37 °C. Thereafter, cells were stimulated with 0.1 ng/mL IL-1β, 50 µM forskolin, 10 µM L-NIL, 10 µM A769662, and 10 µM dorsomorphin dihydrochloride for 24 h. The cells were plated with 8 repetitions for each combination of stimulation. Experiments were performed with a minimum of 5 biological replicates (donors) to ensure reproducibility.

### 4.6. Western Blot

Isolated chondrocytes were plated on 12-well culture plates at a density of 4 × 10^5^ cells/cm^2^. The cells were cultured and stimulated as described above. Chondrocytes were lysed on ice using RIPA lysis buffer (catalog #9806, Cell Signaling, Danvers; MA, USA) freshly supplemented with protease inhibitor cocktail tables (Roche, Mannheim, Germany), transferred in Eppendorf tubes, and frozen at −80 °C. After a freeze-thaw cycle, cell lysates were centrifuged at 15,000× *g* for 10 min to remove cellular debris. Protein extracts were mixed with Laemmli-buffer and separated on 10% polyacrylamide gels at 125 V and transferred to a polyvinylidene difluoride membrane (Millipore, Billerica, MA, USA).

Immunoblotting was performed with antibodies against iNOS (Invitrogen, catalog #PA1-036) and Gapdh (Sigma Aldrich, catalog #G8795) at 4 °C overnight following incubation with HRP-linked secondary antibodies (KPL, Gaithersburg, MD, USA). Protein signals were visualized with a chemiluminescence reaction reagent (catalog #34075, Thermo Scientific, Waltham, MA, USA) according to the manufacturer’s instruction using a CCD camera system (Synoptics, Cambridge, UK). Densitometry of Western blot images was performed using NIH Image J software (available under: https://imagej.nih.gov/ij/, accessed on 1 January 2021).

### 4.7. Statistical Analysis

For statistical analyses, the software SPSS statistics 21 (SPSS, Inc, Chicago, IL, USA) was used. Results were expressed as means ± SEM or percent of control. For multiple sample comparison repeated-measures one-way analysis of variance or Friedman test were performed followed by a paired Student’s test or Wilcoxon signed-rank test, when appropriate. When required, Bonferroni adjustment was performed for multiple comparisons. Significance was accepted at *p* < 0.05.

## 5. Conclusions

The results of the current study show that NO is critically involved in the IL-1β-induced mitochondrial dysfunction in human OA chondrocytes. Inhibition of NO production by iNOS inhibitor, cAMP elevation, or AMPK inhibition prevented the IL-1β-induced negative effects on the mitochondrial function. Thus our study supports the idea that treatment with iNOS inhibitors may be chondroprotective by acting against pathogenic IL-1β-effects on mitochondrial function. 

## Figures and Tables

**Figure 1 ijms-22-02477-f001:**
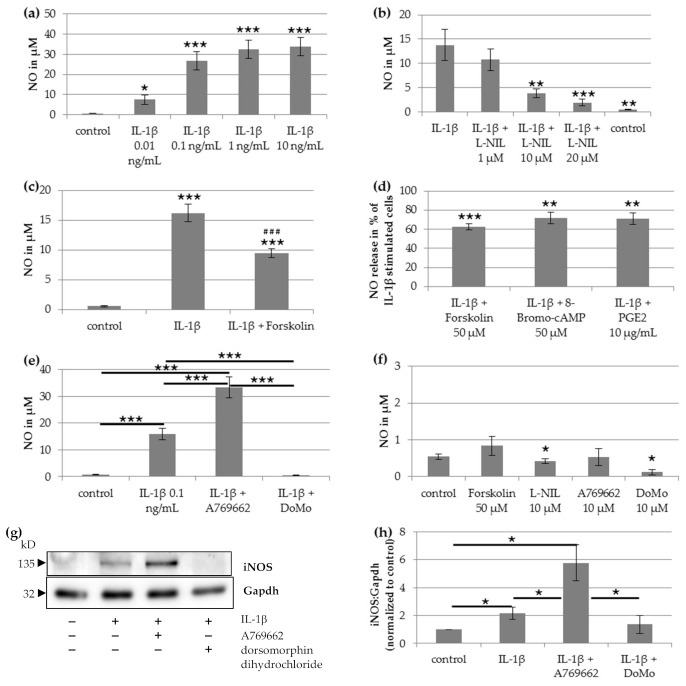
Impact of IL-1β on nitric oxide (NO) release and NO synthase (iNOS) expression of human chondrocytes. (**a**) NO release after stimulation with Interleukin (IL)-1β at different concentrations. Statistical differences vs. unstimulated cells, repeated-measures ANOVA (*n* = 7). (**b**) NO release after stimulation with IL-1β (0.1 ng/mL) and L-NIL at different concentrations. Statistical differences vs. IL-1β-stimulated cells, repeated-measures ANOVA (*n* = 7). (**c**) NO release after stimulation with 0.1 ng/mL IL-1β and 50 µM forskolin. Statistical differences *** vs. unstimulated cells or ^###^ vs. IL-1β-stimulated cells, Wilcoxon signed-rank test and Bonferroni adjustment (*n* = 22). (**d**) NO release after simultaneous stimulation of 0.1 ng/mL IL-1β with 50 µM forskolin (*n* = 22), 50 µM 8-Bromo-cAMP (*n* = 10), or 10 µg/mL PGE_2_ (*n* = 9) normalized to IL-1β-stimulated cells. Statistical differences vs. IL-1β-stimulated cells, Wilcoxon signed-rank test. (**e**) Impact of activation or inhibition of AMPK on IL-1β-induced NO release. Statistic: Repeated-measures ANOVA with Post Hoc test and Bonferroni adjustment (*n* = 9). (**f**) NO release after stimulation with 50 µM forskolin, 10 µM L-NIL, 10 µM A769662, or 10 µM dorsomorphin dihydrochloride (DoMo). Statistical differences vs. unstimulated cells, Wilcoxon signed-rank test (*n* = 5). (**g**) Representative Western blot of iNOS expression after stimulation with/without 0.1 ng/mL IL-1β, 50 µM forskolin, 10 µM A769662 and 10 µM dorsomorphin dihydrochloride. (**h**) Densitometric quantification of iNOS/Gapdh expression after stimulation with 0.1 ng/mL IL-1β, 10 µM A769662 or 10 µM dorsomorphin dihydrochloride normalized to unstimulated cells (*n* = 6). Statistic: Wilcoxon signed-rank test. Values are reported as mean or percentage ± SD. *** *p* < 0.005, ** *p* < 0.01 and * *p* < 0.05. DoMo: Dorsomorphin dihydrochloride.

**Figure 2 ijms-22-02477-f002:**
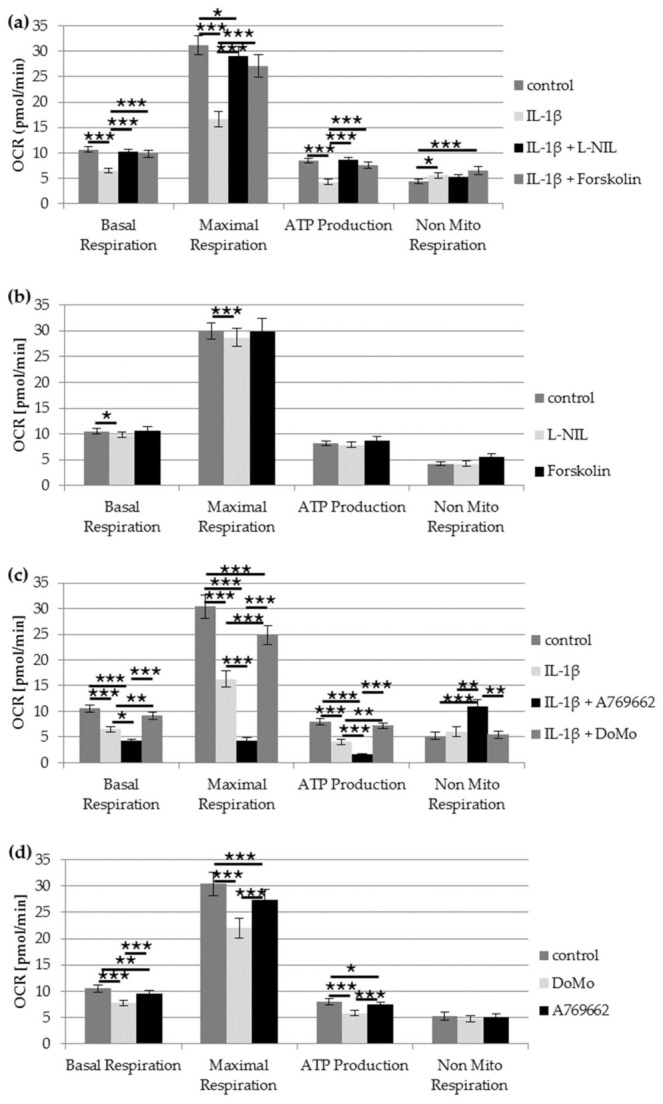
Impact of IL-1β on mitochondrial respiration in chondrocytes. (**a**) Impact of 50 µM forskolin and 10 µM L-NIL (iNOS inhibition) on IL-1β (0.1 ng/mL)-induced mitochondrial dysfunction; *n* = 14. (**b**) Control experiments on the impact of 50 µM forskolin (*n* = 11) and 10 µM L-NIL (*n* = 13) on mitochondrial function. (**c**) Impact of activation or inhibition of AMPK on IL-1β-induced mitochondrial dysfunction (*n* = 12). (**d**) Control experiments on the impact of 10 µM A769662 and 10 µM dorsomorphin dihydrochloride on mitochondrial function (*n* = 12). Statistic: Wilcoxon signed-rang test and Bonferroni adjustment. Values are reported as mean ± SD. *** *p* < 0.005, ** *p* < 0.01 and * *p* < 0.05. OCR: Oxygen consumption rate, DoMo: Dorsomorphin dihydrochloride.

**Figure 3 ijms-22-02477-f003:**
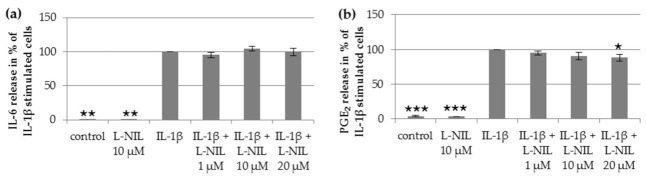
Impact of L-NIL on the IL-1β-induced release of (**a**) IL-6 (*n* = 6), (**b**) PGE_2_ (*n* = 5), (**c**) MMP3 (*n* = 7), and (**d**) the production of GAG (*n* = 6). Values are reported as percentage ± SD normalized to IL-1β-stimulated cells. Statistical differences vs. IL-1β-stimulated cells, repeated-measures ANOVA with Post Hoc test; *** *p* < 0.005, ** *p* < 0.01 and * *p* < 0.05.

**Figure 4 ijms-22-02477-f004:**
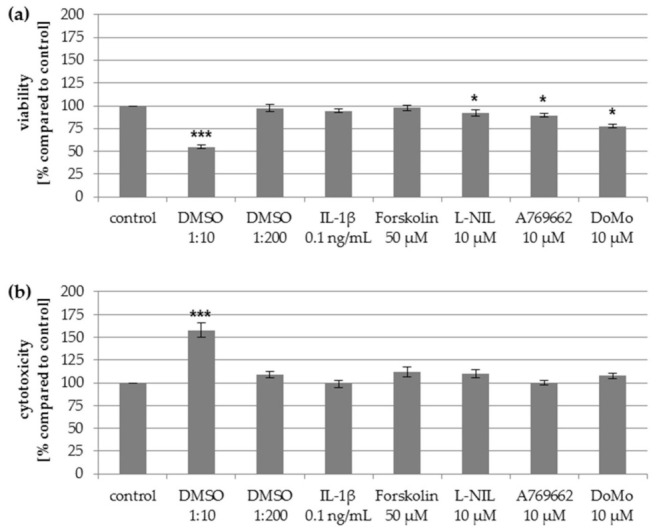
Impact of mediators used on (**a**) viability and (**b**) cytotoxicity of chondrocytes as measured by the LIVE/DEAD viability/cytotoxicity kit, which determines the percentage of living and dead cells. Viability and cytotoxicity values of treated cells are reported as percentage ± SD of control (unstimulated cells), set as 100%. Statistical differences vs. unstimulated cells, Wilcoxon signed-rank test, *** *p* < 0.005 and * *p* < 0.05. DoMo: Dorsomorphin dihydrochloride.

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
