# Peer review of "Inhibition of Inducible Nitric Oxide Synthase Prevents IL-1β-Induced Mitochondrial Dysfunction in Human Chondrocytes"

_ijms, 2021, doi:10.3390/ijms22052477_

Round 1

Reviewer 1 Report

The authors of this paper investigated if blockade of NO production prevents IL-1β- induced mitochondrial dysfunction in chondrocytes and the potential role of cAMP and AMP-activated protein kinase. The role of cytokines in NO production is well known already. However the authors describe a direct link between cAMP and mytochondrial dysfunction.

1-Please include in the introduction more literature related to this topic in relation to cartilage and why there are some gaps in the field of OA that authors aim to answer with their work.

2- Only one concentration for all inhibitors has been used. I am wondering if the authors have used different concentrations too and how are they sure of the real inhibition.  Are there any measurements done in this sense? This should be addressed as a limitation of the study in the disussion.

3- Patient number differs for all studies. Why is this? 

Author Response

1-Please include in the introduction more literature related to this topic in relation to cartilage and why there are some gaps in the field of OA that authors aim to answer with their work.

We revised the introduction, included additional references and specify the aim of the study.

Page 2 line 55:  NO increases the release of pro-inflammatory mediators, inhibits the synthesis of cartilage matrix components, and increases the activity of matrix degrading enzymes such as matrix metalloproteinases (MMP) [10]. Exogenous cytokines can increase iNOS expression and NO release of OA cartilage and cultured OA chondrocytes [11,12]. Page 2 line 61: It has been reported, that mitochondrial dysfunction also increases the responsiveness of chondrocytes to cytokines [14]. An interesting question is therefore, whether NO-induced mitochondrial dysfunction also increases IL-1β-induced release of pro-inflammatory mediators. Page 2 line 79: Whether cAMP and AMPK modulate IL-1β-induced NO release and mitochondrial function in chondrocytes is unclear. Page 2 line 84: Of particular interest is to evaluate whether blockade of iNOS activity prevents the negative effects of IL-1β on chondrocytes. Additionally, we analyzed the role of cAMP and AMPK in the regulation of NO production and release from these cells, and in IL-1β-induced mitochondrial dysfunction. To understand the regulation of iNOS activity in chondrocytes is important to move forward with the development of OA therapies based on iNOS as a target for OA treatment.

2- Only one concentration for all inhibitors has been used. I am wondering if the authors have used different concentrations too and how are they sure of the real inhibition.  Are there any measurements done in this sense? This should be addressed as a limitation of the study in the disussion.

In initial experiments we used different concentrations of mediators and inhibitors/activators and found a concentration-dependent reaction. In the manuscript we presented only the NO release after stimulation with different concentration of IL-1β and L-NIL, because these results are key results of the study. For all other stimulants we chose the lowest effective concentration.  Since the chosen concentrations were also often used in other cell experiments by other authors, we did not include these measurements for all stimulants for reasons of clarity. Furthermore, the initial experiments to choose the best concentration were not always repeated 5 times.

We included in the method section a statement: page11, line 394: After preliminary experiments with different concentrations of all mediators in combination with IL-1β we stimulated the chondrocytes in the final experiments with 50 µM forskolin, 10 µg/ml PGE2, 50 µM 8-Bromo-cAMP, 10 µM L-NIL, 10 µM A769662 (an activator of AMPK) and 10 µM dorsomorphin dihydrochloride (an inhibitor of AMPK) alone or in combination with IL-1β for 48 hours.

And addressed this also as a limitation of the study in the discussion page 10, line 346: Concerning modulation of mitochondrial function by cAMP and AMPK most experiments were performed with a single concentration of the used mediators according to preliminary experiments and literature. The described effects of cAMP and AMPK could be stronger or weaker with other concentrations or incubation times.

3- Patient number differs for all studies. Why is this?

After isolation of chondrocytes from cartilage a limited number of experiments / combination of stimulations was possible with the available material. Therefore, not all experiments could be performed with the cells of every patient. But the stimulation of chondrocytes with IL-1β and IL-1β+Forskolin was necessary as a control in all experiments resulting therefore in a higher patient number compared with the other experiments. 

new references:

  1. Cillero-Pastor, B.; Rego-Pérez, I.; Oreiro, N.; Fernandez-Lopez, C.; Blanco, F. J., Mitochondrial respiratory chain dysfunction modulates metalloproteases -1, -3 and -13 in human normal chondrocytes in culture. BMC musculoskeletal disorders 2013, 14, 235.
  2. Henrotin, Y. E.; Zheng, S. X.; Deby, G. P.; Labasse, A. H.; Crielaard, J. M.; Reginster, J. Y., Nitric oxide downregulates interleukin 1beta (IL-1beta) stimulated IL-6, IL-8, and prostaglandin E2 production by human chondrocytes. J Rheumatol 1998, 25 (8), 1595-601.
  3. Johnson, C. I.; Argyle, D. J.; Clements, D. N., In vitro models for the study of osteoarthritis. The Veterinary Journal 2016, 209, 40-49.
  4. Leonidou, A.; Lepetsos, P.; Mintzas, M.; Kenanidis, E.; Macheras, G.; Tzetis, M.; Potoupnis, M.; Tsiridis, E., Inducible nitric oxide synthase as a target for osteoarthritis treatment. Expert Opinion on Therapeutic Targets 2018, 22 (4), 299-318.
  5. Vuolteenaho, K.; Moilanen, T.; Al-Saffar, N.; Knowles, R. G.; Moilanen, E., Regulation of the nitric oxide production resulting from the glucocorticoid-insensitive expression of iNOS in human osteoarthritic cartilage. Osteoarthritis and cartilage / OARS, Osteoarthritis Research Society 2001, 9 (7), 597-605.
  6. Vuolteenaho, K.; Moilanen, T.; Jalonen, U.; Lahti, A.; Nieminen, R.; van Beuningen, H. M.; van der Kraan, P. M.; Moilanen, E., TGFbeta inhibits IL-1 -induced iNOS expression and NO production in immortalized chondrocytes. Inflamm Res 2005, 54 (10), 420-7.

Reviewer 2 Report

In this manuscript, the authors investigated the possibility to reduce the mitochondtial dysfunction in chondrocytes by blocking the production of NO, as well as the relationship between cAMP or AMPK with NO production. The study was well designed, with clear objectives and appropriate control groups. The results were adequate to meet the objectives.

There are only some minor points that the authors can consider to further improve the manuscript:

  1. Since only in-vitro primary chondrocyte culture was involved in this study, to better justify the validity of this model, the authors should provide some validation tests (protein analysis/physiological function) to justify if the primary cells isolated behaved similarly with the cells in-vivo?
  2. The authors can give a bit more information about the reason why IL-6, PGE2 and MMP3 were being investigated. Also, on page 9, line 302, “It may be therefore, that several pathways converge on mitochondrial respiration chain complexes.” Any examples of these pathways that the authors may suggest and discuss a bit?
  3. Any reason to support the stimulation of chondrocytes with IL-1b for 48 hours but not other period?
  4. In Figure 4b, the expression of the physical meaning of the y-axis is difficult to be understand. Any chance to change the way to present this piece of result?

Author Response

 There are only some minor points that the authors can consider to further improve the manuscript:

  1. Since only in-vitro primary chondrocyte culture was involved in this study, to better justify the validity of this model, the authors should provide some validation tests (protein analysis/physiological function) to justify if the primary cells isolated behaved similarly with the cells in-vivo?

In our presented study, chondrocytes cultured in monolayer were required to perform experiments with the seahorse analyzer and to measure oxygen consumption of mitochondria respiration. For reasons of comparability all experiments were done with isolated chondrocytes cultured in monolayer. To avoid alteration of chondrocytes, isolated cells were cultured only few days and stimulated only 48h.

We know that our approach has some limitations and presumably does not reflect exactly the in-vivo situation. But to study molecular pathways of a specific cell type in-vitro models are essential and often the only possible method. Chondrocytes cultured in monolayer is the most widely used model, and OA-like responses after stimulation with cytokines have been shown with this approach in the literature (see review: Johnson et al. 2016: In vitro models for the study of osteoarthritis).

We included a statement for using this in-vitro model in the limitation section. Page 10, line 350: In general, in-vitro models presumably do not reflect exactly the in-vivo situation in OA cartilage, but chondrocytes cultured in monolayer is the most widely used in-vitro model to study the effect of cytokines on molecular pathways of chondrocytes [28].

  1. The authors can give a bit more information about the reason why IL-6, PGE2 and MMP3 were being investigated. Also, on page 9, line 302, “It may be therefore, that several pathways converge on mitochondrial respiration chain complexes.” Any examples of these pathways that the authors may suggest and discuss a bit?

As described in the discussion section (page 9, line 301) inhibition of mitochondrial respiration complexes increased the expression of cyclooxygenase 2 and the level of PGE2 in normal human chondrocytes as well as the inflammatory responsiveness to cytokines. Besides PGE2, IL-6 and MMP3 are important molecules in OA processes, several studies found that mitochondrial dysfunction affects the production all of these mediators. To point out the importance of these mediators, we revised and specified the whole paragraph, and included additional references in the discussion section.

Page 9 line 309: While we identified significant negative NO effects on mitochondrial function, our data did not provide evidence that NO is critically involved in the IL-1β-induced production and release of IL-6, PGE2 and MMP3. Beside the fact, that PGE2, IL-6 and MMP3 are important molecules in OA processes, several studies found that mitochondrial dysfunction affects the production of these mediators [14,24,25,26]. In human chondrocytes, mitochondrial dysfunction induced by inhibitors of mitochondrial complexes increased the production of PGE2 and MMP3 [25,26] and the inflammatory response to IL-1β [14]. According to these studies, we expected a reduced release of IL-6, PGE2 and MMP3 after incubation with L-NIL and prevention of NO-induced mitochondrial impairment. However, we found, that the mediators PGE2, IL-6 and MMP3  were significantly more released upon stimulation with IL-1β, but the IL-1β-induced release was not affected by L-NIL at concentrations that inhibited the effects of IL-1β on NO production, mitochondrial respiration and ATP production. Thus, while IL-1β alone reduced mitochondrial function in a NO-dependent manner, this effect was not crucial for the IL-1β-induced release of IL-6, PGE2 and MMP3 in our experiments. In the study of Vaamonde-Garcia et al. mitochondrial impairment induced by oligomycin alone already enhanced the basal release of pro-inflammatory mediators, oligomycin combined with IL-1β led to an additional increase of these mediators [14]. Thus the mitochondrial dysfunction was directly initiated by inhibitors additionally to the IL-1β-induced mitochondrial impairment. It seems, that mitochondrial dysfunction induced by inhibitors of mitochondrial respiration chain complexes results in a slightly different reaction compared to the inhibitory effect of NO on mitochondrial respiration. Also anti-inflammatory effects of NO in chondrocytes were described in the literature. In one study, inhibition of NO synthesis enhanced the IL-1β-induced IL-6 and PGE2 production [27]. Concerning the different mentioned effects of NO and inhibitors, several pathways might converge on mitochondrial respiration and result in different responsiveness to cytokines.

  1. Any reason to support the stimulation of chondrocytes with IL-1b for 48 hours but not other period?

Before we started our experiments, we searched the literature concerning incubation time and mediator concentrations. Vuolteenaho et al. 2005 investigated the iNOS expression and NO production in chondrocytes. They described an increase of NO production after IL-1β stimulation up to 48 h follow-up, whereas the highest expression of iNOS was after 24 h. NO measurements on the effect of TGFβ on IL-1β-induced NO releases were always done after 48h. In these experiments with co-stimulation of IL-1β and TGF, they also chose the same IL-1β concentration we chose for our experiments. So we orientated our study design on the experience of other research on chondrocytes.

  1. In Figure 4b, the expression of the physical meaning of the y-axis is difficult to be understand. Any chance to change the way to present this piece of result?

We changed the labeling of the y-axis and the figure legend. Page 7 Figure 4 and page 7 line 221.

new references:

  1. Cillero-Pastor, B.; Rego-Pérez, I.; Oreiro, N.; Fernandez-Lopez, C.; Blanco, F. J., Mitochondrial respiratory chain dysfunction modulates metalloproteases -1, -3 and -13 in human normal chondrocytes in culture. BMC musculoskeletal disorders 2013, 14, 235.
  2. Henrotin, Y. E.; Zheng, S. X.; Deby, G. P.; Labasse, A. H.; Crielaard, J. M.; Reginster, J. Y., Nitric oxide downregulates interleukin 1beta (IL-1beta) stimulated IL-6, IL-8, and prostaglandin E2 production by human chondrocytes. J Rheumatol 1998, 25 (8), 1595-601.
  3. Johnson, C. I.; Argyle, D. J.; Clements, D. N., In vitro models for the study of osteoarthritis. The Veterinary Journal 2016, 209, 40-49.
  4. Leonidou, A.; Lepetsos, P.; Mintzas, M.; Kenanidis, E.; Macheras, G.; Tzetis, M.; Potoupnis, M.; Tsiridis, E., Inducible nitric oxide synthase as a target for osteoarthritis treatment. Expert Opinion on Therapeutic Targets 2018, 22 (4), 299-318.
  5. Vuolteenaho, K.; Moilanen, T.; Al-Saffar, N.; Knowles, R. G.; Moilanen, E., Regulation of the nitric oxide production resulting from the glucocorticoid-insensitive expression of iNOS in human osteoarthritic cartilage. Osteoarthritis and cartilage / OARS, Osteoarthritis Research Society 2001, 9 (7), 597-605.
  6. Vuolteenaho, K.; Moilanen, T.; Jalonen, U.; Lahti, A.; Nieminen, R.; van Beuningen, H. M.; van der Kraan, P. M.; Moilanen, E., TGFbeta inhibits IL-1 -induced iNOS expression and NO production in immortalized chondrocytes. Inflamm Res 2005, 54 (10), 420-7.